# Adaptive federated filter for multi-sensor nonlinear system with cross-correlated noises

**Lijun Wang**[1,2], **Sisi Wang**[1]*, **Wenzhi Yang**[1]

**1** School of Navigation, Guangdong Ocean University, Zhanjiang, China, **2** Hubei Key Laboratory of Inland Shipping Technology, Wuhan, China

☯ These authors contributed equally to this work.
* wangss@gdou.edu.cn

## Abstract

This paper presents an adaptive approach to the federated filter for multi-sensor nonlinear systems with cross-correlations between process noise and local measurement noise. The adaptive Gaussian filter is used as the local filter of the federated filter for the first time, which overcomes the performance degradation caused by the cross-correlated noises. Two kinds of adaptive federated filters are proposed, one uses a de-correlation framework as local filter, and the subfilter of the other one is defined as a Gaussian filter with correlated noises at the same-epoch, and much effort is made to verify the theoretical equivalence of the two algorithms in the nonlinear fusion system. Simulation results show that the proposed algorithms are superior to the traditional federated filter and Gaussian filter with same-paced correlated noises, and the equivalence between the proposed algorithms and high degree cubature federated filter is also demonstrated.

## 1. Introduction

Federated filters (FF) have been successfully used in a wide range of areas, including integrated navigation [1, 2], multi-sensor target tracking [3, 4], gyro-less attitude determination [5], motion capture in virtual reality [6], airborne position & orientation [7] and so on, which have advantages of good real-time, simple structure and high fault-tolerant capability [8]. The original FF proposed by Calson was designed for linear decentralized navigation systems [9, 10]. With the increasing complexity of application systems and environments, many improved FF have been developed for different practical problems. An improved Tobit regression model is applied to the traditional FF framework to form a distributed federated Tobit Kalman filter for censoring and packet delay of a class of discrete time systems [11]. A distributed federated Kalman filter with finite length buffer is proposed to deal with measurement delay or loss for a class of multi-sensor unreliable networked systems with uncorrelated noises [12]. In [1–3, 7, 8, 10–12], the dynamic and measurement models of the system are linear, so the proposed FF adopts the linear Kalman filter (KF) as the local filter. However, the nonlinear problem is often unavoidable in practical systems, and the performance of linear KF is obviously unable to meet the requirements [13–16]. To solve such problems, the information fusion algorithm must be based on the nonlinear fusion mechanism. For example, in order to solve the filtering

**Data Availability Statement:** All relevant data are within the manuscript and its Supporting Information files.

**Funding:** The work was supported in part by Fund of Hubei Key Laboratory of Inland Shipping

Technology under Grant (NHHY2018003); Scientific Research Start-up Funds of Guangdong Ocean University under Grants (E15031, R17012); Characteristic Innovation Projects of Guangdong Province under Grants (2017KTSCX088, 2017KTSCX092, 2019KTSCX230). There was no additional external funding received for this study.

**Competing interests:** The authors have declared that no competing interests exist.

precision decline and instability problem caused by various factors in the practical application of the integrated navigation system based on the nonlinear model, [17–20] proposed several solutions based on the unscented Kalman filter. Therefore, it is necessary to adopt nonlinear filter as local filter to study federated filter. Different federated UKF are designed for pulsar/ CNS satellite integrated autonomous navigation system and vehicle fusion positioning system [21, 22]. A federated nonlinear predictive filtering method is proposed for the gyroless attitude determination system with star sensors and GPS sensors, which combines the nonlinear predictive filtering with the traditional FF [5].

Measurement noise and process noise in existing nonlinear federated filters (NFF) are generally considered to be statistically independent [4, 8, 21]. However, according to the research experience of a single sensor system, the cross-correlation between process noise and measurement noise is an important factor that leads to the degradation of filtering performance, which also exists in multi-sensor system. In practical application, the problem of noise cross-correlation always exists. For example, in a target tracking system, there would be some cross-correlation between the process and measurement noises if both of them are dependent on the system state [23]. Also, discretization on real continuous fusion systems may cause the cross-correlation to the process and measurement noises [24]. Several fusion filtering algorithms have been brought up to alleviate the negative effects of noise correlation in multi-sensor systems. For a class of uncertain multi-sensor systems with autocorrelation and cross-correlation noises, a distributed weighted robust fusion filter is constructed by using the optimal robust Kalman local filter [23]. The problem of information fusion estimation for multi-sensor stochastic uncertain systems with correlated noises is presented in [24], in which the process and observation noises are one-step auto-correlated and two-step cross-correlated respectively, while the observation noises of different sensors are one-step cross-correlated. Based on the optimal local filter in [25], a distributed fusion filter for multi-sensor systems with finite-step correlated noises is proposed, in which process noise and observation noise at different sensors are finite-step auto-correlated and cross-correlated respectively [26]. Decentralized cubic Kalman fusion filters are proposed for nonlinear fusion systems with one-step cross-correlations between the process noise and measurement noise and synchronized cross-correlations among each measurement noise [27]. However, it can be seen from [23–27] that the aforementioned algorithms are designed for stochastic uncertain systems with cross-correlated noise or auto-correlated noise. The time-varying state transition matrix and measurement matrix in these system equations are linear with the system state, so these improved fusion algorithms still belong to the category of linear fusion algorithms in a sense. The fusion algorithm proposed in [27] is designed for nonlinear multi-sensor system, but it is mainly used to overcome the one-step correlation between the process and measurement noises, and the correlations between different sensor noises. To the best of the authors' knowledge, few studies have been done on multi-sensor systems with the same-paced cross-correlation between process and measurement noise.

In a single sensor system, the problem of filtering under non-standard noise has always been the focus of researchers. An improved adaptive student's t-filter for the filtering of the linear system in the context of independent non-Gaussian heavy-tailed noise was proposed in [28–30]. A pseudo-measurement noise was constructed to form a pseudo-observation equation for the filtering of one-step related noise in nonlinear systems, and on this basis, new Gaussian approximation filters and smoothers are derived [31]. A novel adaptive Kalman filter based on Variational Bayesian method and Gauss-Inverse-Wishart mixture distribution was proposed for the linear system filtering problem with unknown system state and observed noise covariance matrix [32]. Reference [33] further improved the above filter based on the approximation of slide Window State Vectors based on the work in reference [32]. Nonlinear

filtering under cross-correlation noise has become an important branch of filtering under non-standard noise. There are two main solutions, one is de-correlating the noise sequences by reconstructing a new pseudo process noise sequence and a new process function [34–36] the other method is to use a Gaussian approximation recursive filter (GASF) for same-paced cross-correlated noise, which adopts the Gaussian approximation to the two-step state posterior predictive probability density function (PDF) and the one-step measurement posterior predictive PDF [34]. The equivalence of the above two methods in linear and nonlinear systems has been proved theoretically [35, 36], but there is a narrow performance gap between them [37]. It should be noted that the performance of these two filters is not satisfactory when applied to nonlinear systems. Accordingly, another general framework of the correlated Gaussian approximated filter (CGAF) for same-paced correlated noises is established by introducing the Gaussian approximation of the conditional PDF of the process noises [38].

Given the feasibility of the above solutions in single-sensor systems, it can be inferred that their applications in traditional NFF frameworks should have similar effectiveness. However, until now, only GASF has been successfully used as a local filter of high degree cubature federated filter for cross-correlated noises (HCFF-CN) [39]. Therefore, it's worth discussing whether the de-correlating filter and CGAF are suitable to modify the NFF. Furthermore, if they are appropriate, what is the difference between their applications? Inspired by previous research and above problems, two improved adaptive federated filters for cross-correlated noises (AFF-CN) are proposed.

The rest of the article is organized as follows: Section 2 formulates the investigated problem. Section 3 is devoted to systematic procedures for two kinds of AFF-CN. Section 4 provides theoretical equivalence of the two algorithms in the nonlinear fusion system. In Section 5, simulation results and discussion are presented. Section 6 draws the conclusion.

## 2. Problem formulation

Considering a class of nonlinear discrete-time stochastic systems with multi-sensors

$$\begin{cases} \boldsymbol{x}_{k+1} = f_k(\boldsymbol{x}_k) + \boldsymbol{w}_k \\ \boldsymbol{z}_{m,k} = h_{m,k}(\boldsymbol{x}_k) + \boldsymbol{v}_{m,k} \end{cases} \tag{1}$$

where $k$ is the discrete sample time index; $m = 1,2,\ldots N$ is the sensor index; $\boldsymbol{x}_k \in \mathbf{R}^n$ and $\boldsymbol{z}_{m,k} \in \mathbf{R}^{P_m}$ are the system state and the $mth$ measurement vector at $k$ respectively, where the superscript $n$ is the dimension of the state vector, $P_m$ is the observation vector dimension of the $m$ sensor; $f_k(\cdot)$ and $h_{m,k}(\cdot)$ are the nonlinear process function and the measurement function of the $mth$ sensor at $k$ respectively; the process noise $\{\boldsymbol{w}_k\}$ and measurement noise $\{\boldsymbol{v}_{m,k}\}$ are cross-correlated zero-mean Gaussian white noise sequences satisfying $\mathrm{E}[\boldsymbol{w}_k \boldsymbol{w}_j^{\mathrm{T}}] = \boldsymbol{Q}_k \delta_{kj}$, $\mathrm{E}[\boldsymbol{v}_{m,k} \boldsymbol{v}_{m,l}^{\mathrm{T}}] = \boldsymbol{R}_{m,k} \delta_{kl}$ and $\mathrm{E}[\boldsymbol{w}_k \boldsymbol{v}_{m,l}^{\mathrm{T}}] = \boldsymbol{D}_{m,k} \delta_{kl}$, where E denotes the mathematical expectation and $\delta_{kl}$ is Kronecker delta function. $\mathrm{E}[\boldsymbol{w}_k \boldsymbol{v}_{m,l}^{\mathrm{T}}] = \boldsymbol{D}_{m,k} \delta_{kl}$ indicates the cross-correlation between $\boldsymbol{w}_k$ and $\boldsymbol{v}_{m,k}$, only if the time indexes are the same. Note that the cross-correlation noises for the rest of the paper are the same as above. The initial state $\boldsymbol{x}_0$ described by the Gaussian distribution is uncorrelated with $\{\boldsymbol{w}_k\}$ and $\{\boldsymbol{v}_{m,k}\}$, and its associated mean and covariance are defined as $\hat{\boldsymbol{x}}_{0|0}$ and $P_{0|0}$.

**Assumption 1:** The $f_k(\cdot)$ and $h_{m,k}(\cdot)$ are known, and the state $x_k$ is bounded.

As mentioned above, the traditional NFF method is not suitable for the nonlinear fusion system described in Eq 1, so an evolutionary filter HCFF-CN is proposed to solve such problem. However, there are still some defects that limit its further application.

**Assumption 2:** The previous measurements satisfy $Z_{m,k-1} = \{z_{m,i}, i = 1, \cdots, k\text{-}1\}$. Both the two-step predictive conditional PDF of the state $p(x_{k+1}|Z_{m,k-1})$ and the one step predictive conditional PDF of the measurements $p(z_k|Z_{m,k-1})$ conform to the Gaussian distribution. HCFF-CN inherits a disadvantage from cubature Kalman filter (CKF), that is, it is easy to lose the positive definiteness of system state error covariance in its cycle steps and make the filter stop running continuously.

Considering the above limitations and assumptions, two kinds of AFF-CN are proposed to supplement and extend HCFF-CN.

## 3. Design of AFF-CN

In general, the nonlinear filter assumes that the state, process and measurement noise all conform to the Gaussian distribution, and the statistical characteristics of the corresponding Gaussian PDFs are obtained by using the multivariate Gauss integral. Because Gaussian integrals are often unanalytical, some point-based numerical rules are applied to approximate these integrals. So The Gaussian integrals with respect to the Gaussian density function are usually approximated as follows [40, 41].

$$\int_{\mathbf{R}^n} m(x)\mathrm{N}(x; \hat{x}, P)dx \approx \sum_{i=1}^{B} W_i m(s\Lambda_i + \hat{x}) \tag{2}$$

where $m(x)$ is a known nonlinear function; $\mathrm{N}(x; \hat{x}, P)$ is a Gaussian density function with mean $\hat{x}$ and covariance $P(P = ss^{\mathrm{T}}$, and $s$ can be obtained by Cholesky decomposition or singular value decomposition); and $B$ is the number of all points; $\Lambda_i$ and $W_i$ are the point generator and weight at $i$, which can be directly obtained based on the corresponding numerical rule. For example, in the 3rd degree cubature rule, $W_i$ and $\Lambda_i$ are defined as follows [40].

$$\begin{cases} W_i = 1/2n, i = 1, 2, \cdots, 2n \\ \Lambda_i = \sqrt{n}e_i, \ i = 1, 2, \cdots, n \\ \Lambda_i = -\sqrt{n}e_{i-n}, \ i = n+1, n+2, \cdots, 2n \end{cases} \tag{3}$$

where $e_i \in \mathrm{R}^n$ is a unit vector with the *ith* element being l, and $n$ is the system state dimension. $B$ is set to be $2n$.

According to Eqs 2 and 3, introducing different point-based numerical rules in the framework of the Gaussian filter leads to different nonlinear Gaussian approximation filters. For instance, the unscented transformation (UT) gives rise to the unscented Kalman filter (UKF), and the Gauss–Hermite quadrature rule brings about the Gauss–Hermite Kalman filter [42, 43]. Accordingly, a special kind of single Gaussian filter (SGF), called point-based nonlinear filter (PNF), is established. Each PNF has its own usage. For example, UKF is applicable to low dimensions systems ($n \leq 3$), while the 3rd degree CKF is suitable for higher dimension systems (n>3) [44].

In order to make the proposed filter have all the advantages of PNF, one derives the point-based AFF-CN, whose local filters are expressed as the universal framework of the PNF. Therefore, AFF-CN can select the most appropriate point-based numerical approximation rule according to the actual application. Later in this section, the general framework of AFF-CN is firstly described, and then, by virtue of the de-correlating filter and CGAF, two kinds of local filters are derived respectively. Furthermore, the algorithm with de-correlating filter as local filter is labeled AFF1-CN, and the other one using CGAF as subfilter is named AFF2-CN.

### 3.1 General framework of AFF-CN

The general architecture of the AFF-CN is similar to that of the traditional FF except for their local filters. To counteract the negative impact on the global approximated optimality caused by the nonlinear local filter, the fusion-reset mode is still used. The procedure of the AFF-CN can be described as the following steps.

*Step 1*: Information Distribution.

$$
\begin{cases}
\boldsymbol{Q}_{i,\,k} = \beta_i^{-1} \boldsymbol{Q}_k \\
\boldsymbol{P}_{i,\,k|k} = \beta_i^{-1} \boldsymbol{P}_{k|k}^g \\
\hat{\boldsymbol{x}}_{i,\,k|k} = \hat{\boldsymbol{x}}_{k|k}^g, \quad i = 1, 2, \ldots, N, M
\end{cases}
\tag{4}
$$

where $\beta_i = 1/N$ and $\beta_M = 0$ are the information distribution coefficients of each local filter and the master filter in fusion-reset mode, respectively, and satisfy $\sum_{i=1}^{N} \beta_i + \beta_M = 1$. The values of $\beta_i$ and $\beta_M$ are the same as those in [10]. $\hat{\boldsymbol{x}}_{i,k|k}$ is the local state estimate, and its associated covariance is $P_{i,k|k}$. $\hat{\boldsymbol{x}}_{k|k}^g$ is the global state estimate and its associated covariance is $\boldsymbol{P}_{k|k}^g$. The information is distributed between the local filters and the master filter according to the above coefficients.

*Step 2*: *Information Update*.

To account for the value of $\beta_i$, the information update step, which consists of prediction and update steps, is performed in each local filter. Through this step, $\hat{\boldsymbol{x}}_{i,k|k}$ and $P_{i,k|k}$ are updated to $\hat{\boldsymbol{x}}_{i,k+1|k+1}$ and $\boldsymbol{P}_{i,k+1|k+1}$.

Step 3: Information Fusion.

Since $\beta_M$ equals 0 and $\beta_M^{-1} \boldsymbol{Q}_k$ approaches infinity, there is no information assigned to the master filter. The global state and its associated covariance are generated as follows

$$
\boldsymbol{P}_{k+1|k+1}^g = \left[ \sum_{i=1}^{N} \boldsymbol{P}_{i,k+1|k+1}^{-1} \right]^{-1}
$$
$$
\hat{\boldsymbol{x}}_{k+1|k+1}^g = \boldsymbol{P}_{k+1|k+1}^g \left[ \sum_{i=1}^{N} \boldsymbol{P}_{i,k+1|k+1}^{-1} \hat{\boldsymbol{x}}_{i,k+1|k+1} \right]
\tag{5}
$$

Through the above recursive cycling steps, the approximated global optimal solution of system (1) is obtained.

### 3.2 Point-based local filter for AFF-CN

In system (1), since $\boldsymbol{w}_k$ is correlated with $\boldsymbol{v}_{m,k}$, and is indirectly correlated with $\boldsymbol{z}_{m,k}$ or $Z_{m,k}$, $\mathrm{E}[\boldsymbol{w}_k|\boldsymbol{Z}_{m,k}] \neq \mathrm{E}[\boldsymbol{w}_k] = 0$. Accordingly, the local state prediction estimates $\hat{\boldsymbol{x}}_{m,k+1|k}$ and its error covariance $P_{m,k+1|k}$ can't be updated as following.

$$
\hat{\boldsymbol{x}}_{m,k+1|k} = \mathrm{E}[f_k(\boldsymbol{x}_k)|\boldsymbol{Z}_{m,k}] + \mathrm{E}[\boldsymbol{w}_k|\boldsymbol{Z}_{m,k}]
$$
$$
\neq \int f_k(\boldsymbol{x}_k) \mathrm{N}(\boldsymbol{x}_k; \boldsymbol{x}_{k|k}, \boldsymbol{P}_{k|k}) d\boldsymbol{x}_k
$$
$$
\boldsymbol{P}_{m,k+1|k} = \mathrm{E}[\tilde{\boldsymbol{x}}_{m,k+1/k} \tilde{\boldsymbol{x}}_{m,k+1|k}^{\mathrm{T}}|\boldsymbol{Z}_{m,k}]
$$
$$
\neq \int f_k(\boldsymbol{x}_k)[f_k(\boldsymbol{x}_k)]^T \mathrm{N}(\boldsymbol{x}_k; \boldsymbol{x}_{k|k}, \boldsymbol{P}_{k|k}) d\boldsymbol{x}_k - \tilde{\boldsymbol{x}}_{k+1/k} \tilde{\boldsymbol{x}}_{k+1|k}^{\mathrm{T}} + \boldsymbol{Q}_k
\tag{6}
$$

Therefore, AFF1-CN and AFF2-CN adopt different methods to maintain the information update procedure in their local filters. The details are set out below.

**3.2.1 Point-based local filter for AFF1-CN.** According to the observation equation in Eq 1 $z_{m,k} = \mathbf{h}_{m,k}(\boldsymbol{x}_k)+\boldsymbol{v}_{m,k}$, one can get $z_{m,k}-\mathbf{h}_{m,k}(\boldsymbol{x}_k)+\boldsymbol{v}_{m,k} = 0$, and then $\boldsymbol{D}_{m,k}\boldsymbol{R}_{m,k}^{-1}[\boldsymbol{z}_{m,k} - \mathbf{h}_{m,k}(\boldsymbol{x}_k) - \boldsymbol{v}_{m,k}]=0$. Obviously, the substitution of this additional term into the process equation of Eq (1) will not change the original equation relationship in the process equation. By introducing an additional term $\boldsymbol{D}_{m,k}\boldsymbol{R}_{m,k}^{-1}[\boldsymbol{z}_{m,k} - h_{m,k}(\boldsymbol{x}_k) - \boldsymbol{v}_{m,k}]$, the state equation in system (1) can be rewritten as follows.

$$
\begin{aligned}
\boldsymbol{x}_{k+1} &= f_k(\boldsymbol{x}_k) + \boldsymbol{w}_k + \boldsymbol{D}_{m,k}\boldsymbol{R}_{m,k}^{-1}\left[\boldsymbol{z}_{m,k} - h_{m,k}(\boldsymbol{x}_k) - \boldsymbol{v}_{m,k}\right] \\
&= f_k(\boldsymbol{x}_k) + \boldsymbol{D}_{m,k}\boldsymbol{R}_{m,k}^{-1}\left[\boldsymbol{z}_{m,k} - h_{m,k}(\boldsymbol{x}_k)\right] + \boldsymbol{w}_k - \boldsymbol{D}_{m,k}\boldsymbol{R}_{m,k}^{-1}\boldsymbol{v}_{m,k}
\end{aligned}
\tag{7}
$$

Define $F_{m,k}(\boldsymbol{x}_k) = f_k(\boldsymbol{x}_k) + \boldsymbol{D}_{m,k}\boldsymbol{R}_{m,k}^{-1}(\boldsymbol{z}_{m,k} - h_{m,k}(\boldsymbol{x}_k))$ and $\widehat{\boldsymbol{w}}_{m,k} = \boldsymbol{w}_k - \boldsymbol{D}_{m,k}\boldsymbol{R}_{m,k}^{-1}\boldsymbol{v}_{m,k}$, so Eq 7 is rewritten as follows

$$
\boldsymbol{x}_{k+1} = F_{m,k}(\boldsymbol{x}_k) + \widehat{\boldsymbol{w}}_{m,k}
\tag{8}
$$

where $\widehat{\boldsymbol{w}}_{m,k}$ satisfies $\mathrm{E}(\widehat{\boldsymbol{w}}_{m,k}\widehat{\boldsymbol{w}}_{m,j}^{\mathrm{T}}) = (\boldsymbol{Q}_k - \boldsymbol{D}_{m,k}\boldsymbol{R}_{m,k}^{-1}\boldsymbol{D}_{m,k}^{\mathrm{T}})\boldsymbol{\delta}_{kj}$, $\mathrm{E}(\widehat{\boldsymbol{w}}_{m,k}\boldsymbol{v}_{m,j}^{\mathrm{T}}) = 0$, and $\mathrm{E}(\widehat{\boldsymbol{w}}_{m,k}) = 0$. Therefore, $\widehat{\boldsymbol{w}}_{m,k}$ is zero-mean Gaussian white noise and uncorrelated with $\boldsymbol{v}_{m,k}$. The fusion system is reconstructed by Eq 8 and the measurement equation of Eq 1, and the problem is transformed into a standard Gaussian filtering problem. The local filters of AFF1-CN can be summarized as follows.

(1) The mean and covariance of the initial state $\boldsymbol{x}_0$ are known.

(2) Evaluate the predicted state $\hat{\boldsymbol{x}}_{m,\,k+1|k}$ and its associated covariance $P_{m,k+1|k}$.

$$
\begin{aligned}
\hat{\boldsymbol{x}}_{m,k+1|k} &= \int_{\mathbf{R}^n} \mathrm{F}_{m,k}(\boldsymbol{x}_k)\mathrm{N}(\boldsymbol{x}_k; \hat{\boldsymbol{x}}_{m,\,k|k}, \boldsymbol{P}_{m,\,k|k})\mathrm{d}\boldsymbol{x}_k \\
&= \sum_{i=1}^{B} W_i \mathrm{F}_{m,k}(\boldsymbol{\xi}_{m,i,k|k})
\end{aligned}
\tag{9}
$$

$$
\begin{aligned}
\boldsymbol{P}_{m,k+1|k} &= \int_{\mathbf{R}^n} (F_{m,k}(\boldsymbol{x}_k) - \hat{\boldsymbol{x}}_{m,k+1|k})(F_{m,k}(\boldsymbol{x}_k) - \hat{\boldsymbol{x}}_{m,k+1|k})^{\mathrm{T}}\mathrm{N}(\boldsymbol{x}_k; \hat{\boldsymbol{x}}_{m,k|k}, \boldsymbol{P}_{m,k|k})\mathrm{d}\boldsymbol{x}_k + \boldsymbol{Q}_k - \boldsymbol{D}_{m,k}\boldsymbol{R}_{m,k}^{-1}\boldsymbol{D}_{m,k}^{\mathrm{T}} \\
&= \sum_{i=1}^{B} W_i (F_{m,k}(\boldsymbol{\xi}_{m,i,k|k}) - \hat{\boldsymbol{x}}_{m,k+1|k})(F_{m,k}(\boldsymbol{\xi}_{m,i,k|k}) - \hat{\boldsymbol{x}}_{m,k+1|k}) + \boldsymbol{Q}_k - \boldsymbol{D}_{m,k}\boldsymbol{R}_{m,k}^{-1}\boldsymbol{D}_{m,k}^{\mathrm{T}}
\end{aligned}
\tag{10}
$$

The transformed points in Eqs 9 and 10 are defined as follows.

$$
\boldsymbol{\xi}_{m,i,k|k} = \boldsymbol{S}_{m,k|k}\boldsymbol{\Lambda}_i + \hat{\boldsymbol{x}}_{m,k|k}
\tag{11}
$$

where $\boldsymbol{S}_{m,k|k}\boldsymbol{S}_{m,k|k}^{\mathrm{T}} = \boldsymbol{P}_{m,k|k}$, and $s_{m,k|k}$ can be obtained in the same way as $s$ in Eq 2.

(3) Update

Update the mean and covariance of $\boldsymbol{x}_{k+1}$.

$$
\begin{aligned}
\hat{\boldsymbol{x}}_{m,k+1|k+1} &= \hat{\boldsymbol{x}}_{m,k+1|k} + \boldsymbol{K}_{m,k+1}(\boldsymbol{z}_{m,k+1} - \hat{\boldsymbol{z}}_{m,k+1|k}) \\
\boldsymbol{P}_{m,k+1|k+1} &= \boldsymbol{P}_{m,k+1|k} - \boldsymbol{K}_{m,k+1}\boldsymbol{P}_{m,k+1|k}^{zz}\boldsymbol{K}_{m,k+1}^{\mathrm{T}}
\end{aligned}
\tag{12}
$$

where the filtering gain $\boldsymbol{K}_{m,k+1}=\boldsymbol{P}_{m,k+1|k}^{xz}(\boldsymbol{P}_{m,k+1|k}^{zz})^{-1}$, and the predicted measurement $\hat{\boldsymbol{z}}_{m,k+1|k}$, the

innovation covariance $P^{zz}_{m,k+1|k}$ and the cross-covariance $P^{xz}_{m,k+1|k}$ are given by

$$\hat{z}_{m,k+1|k} = \int_{\mathbf{R}^n} h_{m,k+1}(x_{k+1})\mathrm{N}(x_{k+1};\hat{x}_{m,\ k+1|k}, P_{m,\ k+1|k})\mathrm{d}x_{k+1}$$

$$=\sum_{i=1}^{B}W_i h_{m,k+1}(\xi_{m,i,k+1|k}) \tag{13}$$

$$P^{zz}_{m,k+1|k} = \int_{\mathbf{R}^n} (h_{m,k+1}(x_{k+1}) - \hat{z}_{m,k+1|k})(h_{m,k+1}(x_{k+1}) - \hat{z}_{m,k+1|k})^{\mathrm{T}} \times \mathrm{N}(x_{k+1};\hat{x}_{m,k+1|k}, P_{m,k+1|k})\mathrm{d}x_{k+1} + R_{m,k+1}$$

$$= \sum_{i=1}^{B}W_i(h_{m,k+1}(\xi_{m,i,k+1|k}) - \hat{z}_{m,k+1|k})(h_{m,k+1}(\xi_{m,i,k+1|k}) - \hat{z}_{m,k+1|k})^{\mathrm{T}} + R_{m,k+1} \tag{14}$$

$$P^{xz}_{m,k+1|k} = \int_{\mathbf{R}^n} (x_{k+1} - \hat{x}_{m,k+1|k})(h_{m,k+1}(x_{k+1}) - \hat{z}_{m,k+1|k})^{\mathrm{T}} \times \mathrm{N}(x_{k+1};\hat{x}_{m,k+1|k}, P_{m,k+1|k})\mathrm{d}x_{k+1}$$

$$= \sum_{i=1}^{B}W_i(\xi_{m,i,k+1|k} - \hat{x}_{m,k+1|k})(h_{m,k+1}(\xi_{m,i,k+1|k}) - \hat{z}_{m,k+1|k})^{\mathrm{T}} \tag{15}$$

where the propagated points in Eq 13 are defined as follows

$$\xi_{m,i,k+1|k} = S_{m,k+1|k}\Lambda_i + \hat{x}_{m,k+1|k} \tag{16}$$

where $S_{m,k+1|k}S^{\mathrm{T}}_{m,k+1|k} = P_{m,k+1|k}$, and $S_{m,k+1|k}$ can be obtained in the same way as $s_{m,k|k}$.

**3.2.2 Point-based local filter for AFF2-CN.** The process noises $w_k$ in Eq 1 is the zero-mean Gaussian white noise with covariance $Q_k$ and uncorrelated with $Z_{m,k-1}$, the conditional PDF $p(w_k|Z_{m,k})$ of $w_k$ is presumed to follow the Gaussian distribution, which is defined as follows

$$p(w_k|Z_{m,k}) \approx \mathrm{N}(w_k;\quad \hat{w}_{m,k|k},\quad P^{ww}_{m,k|k}) \tag{17}$$

where the mean $\hat{w}_{m,k|k}$ and associated error covariance $P^{ww}_{m,k|k}$ can be estimated as follows

$$\begin{cases} \hat{w}_{m,k|k} = K^w_{m,k}(z_{m,k} - \hat{z}_{m,k|k-1}) \\ P^{ww}_{m,k|k} = Q_k - K^w_{m,k}P^{zz}_{m,k|k-1}(K^w_{m,k})^{\mathrm{T}} \\ K^w_{m,k} = D_{m,k}(P^{zz}_{m,k|k-1})^{-1} \end{cases} \tag{18}$$

where, $\hat{z}_{m,k|k-1}$ and $P^{zz}_{m,k|k-1}$ are as the same as those in Eqs 13 and 14. Based on the above assumption, the point-based local filters for AFF2-CN are formulated as follows

(1) The mean and covariance of the initial state $x_0$ are known.

(2) Evaluate the predicted state $\hat{\boldsymbol{x}}_{m,\ k+1|k}$ and $\boldsymbol{P}_{m,k+1|k}$.

$$\hat{\boldsymbol{x}}_{m,\ k+1|k} = \int_{\mathbf{R}^n} f_k(\boldsymbol{x}_k) \mathrm{N}(\boldsymbol{x}_k; \hat{\boldsymbol{x}}_{m,\ k|k}, \boldsymbol{P}_{m,\ k|k}) \mathrm{d}\boldsymbol{x}_k + \hat{\boldsymbol{w}}_{m,k|k}$$
$$= \sum_{i=1}^{B} W_i f_k(\boldsymbol{\xi}_{m,i,k|k}) + \hat{\boldsymbol{w}}_{m,k|k} \tag{19}$$

$$\boldsymbol{P}_{m,\ k+1|k} = \int_{\mathbf{R}^n} \left[f_k(\boldsymbol{x}_k) + \hat{\boldsymbol{w}}_{m,x,k|k}\right] \left[f_k(\boldsymbol{x}_k) + \hat{\boldsymbol{w}}_{m,x,k|k}\right]^{\mathrm{T}} \times \mathrm{N}(\boldsymbol{x}_k; \hat{\boldsymbol{x}}_{m,\ k|k}, \boldsymbol{P}_{m,\ k|k}) \mathrm{d}\boldsymbol{x}_k - \hat{\boldsymbol{x}}_{m,k+1|k}\hat{\boldsymbol{x}}_{m,k+1|k}^{\mathrm{T}} + \boldsymbol{\Omega}_{m,k|k}$$
$$= \sum_{i=1}^{B} W_i \left[f_k(\boldsymbol{\xi}_{m,i,k|k}) + \hat{\boldsymbol{w}}_{m,x,k|k}\right] \left[f_k(\boldsymbol{\xi}_{m,i,k|k}) + \hat{\boldsymbol{w}}_{m,x,k|k}\right]^{\mathrm{T}} - \hat{\boldsymbol{x}}_{m,k+1|k}\hat{\boldsymbol{x}}_{m,k+1|k}^{\mathrm{T}} + \boldsymbol{\Omega}_{m,k|k} \tag{20}$$

where

$$\begin{cases} \hat{\boldsymbol{w}}_{m,x,k|k} = \hat{\boldsymbol{w}}_{m,k|k} + (\boldsymbol{P}_{m,k|k}^{xw})^{\mathrm{T}} \boldsymbol{P}_{m,k|k}^{-1} (\boldsymbol{\xi}_{m,i,k|k} - \hat{\boldsymbol{x}}_{m,k|k}) \\ \boldsymbol{\Omega}_{m,k|k} = \boldsymbol{P}_{m,k|k}^{ww} - (\boldsymbol{P}_{m,k|k}^{xw})^{\mathrm{T}} \boldsymbol{P}_{m,k|k}^{-1} \boldsymbol{P}_{m,k|k}^{xw} \end{cases} \tag{21}$$

where $\boldsymbol{P}_{m,k|k}^{xw} = -\boldsymbol{P}_{m,k|k-1}^{xz}(\boldsymbol{P}_{m,k|k-1}^{zz})^{-1}\boldsymbol{D}_{m,k}^{\mathrm{T}}$. $W_i$ and $\boldsymbol{\xi}_{m,i,k|k}$ are the same as those in Eq 9.

(3) Update

The updated step is identical with that in the local filters of AFF1-CN.

In summary, before local filtering, the AFF1-CN de-correlates the process and measurement noises. And in the AFF2-CN, the posterior PDF of $\boldsymbol{w}_k$ is considered to follow the Gaussian distribution, and the maximum a posteriori estimate (MAP) is used to estimate $\boldsymbol{w}_k$. Therefore, two point-based AFF-CN are established, which can introduce any suitable point-based numerical approximation rules.

## 4. Equivalence proof of AFF1-CN and AFF2-CN

The local filters of AFF1-CN and HCFF-CN use the de-correlating filtering framework and the correlated recursive Gaussian approximated filtering framework, respectively [34, 39]. In [35, 36], the theoretical equivalence had already been proved between the de-correlating filtering framework and GASF for linear and nonlinear systems, which means that AFF1-CN is equivalent to HCFF-CN when it is approximated for the five-degree cubature rule. Can it be inferred that AFF1-CN and AFF2-CN also have the theoretical equivalence? This is a problem to be solved in this section. Obviously, from Eqs 4–21, the only difference existing between AFF1-CN and AFF2-CN is the estimation computation of the predicted states $\hat{\boldsymbol{x}}_{m,\ k+1|k}$ and its associated error covariance $\boldsymbol{P}_{m,\ k+1|k}$, which are expressed as Eqs 9, 10, 19, 20 respectively. Therefore, the following verification of equivalence focuses on the argument that those Eqs 9 and 10 are equivalent to the Eqs and 20.

**Proof.** According to the definition of $F_{m,k}(\boldsymbol{x}_k)$ in Eqs 8 and 9 is expanded as follows

$$\hat{\boldsymbol{x}}_{m,k+1|k} = \int_{\mathbf{R}^n} f_k(\boldsymbol{x}_k) \mathrm{N}(\boldsymbol{x}_k; \hat{\boldsymbol{x}}_{m,\ k|k}, \boldsymbol{P}_{m,\ k|k}) \mathrm{d}\boldsymbol{x}_k + \boldsymbol{D}_{m,k} \boldsymbol{R}_{m,k}^{-1} (\boldsymbol{z}_{m,k} - \hat{\boldsymbol{z}}_{m,k|k}) \tag{22}$$

where $\hat{\boldsymbol{z}}_{m,k|k} = \int_{\mathbf{R}^n} h_{m,k}(\boldsymbol{x}_k) \mathrm{N}(\boldsymbol{x}_k; \hat{\boldsymbol{x}}_{m,\ k|k}, \boldsymbol{P}_{m,\ k|k}) \mathrm{d}\boldsymbol{x}_k$.

Substitute Eq 18 into Eq 19, one can get

$$\hat{\boldsymbol{x}}_{m,\,k+1|k} = \int_{\mathbf{R}^n} f_k(\boldsymbol{x}_k)\mathrm{N}(\boldsymbol{x}_k; \hat{\boldsymbol{x}}_{m,\,k|k}, \boldsymbol{P}_{m,\,k|k})\mathrm{d}\boldsymbol{x}_k + \boldsymbol{D}_{m,k}(\boldsymbol{P}_{m,k|k-1}^{zz})^{-1}(\boldsymbol{z}_{m,k} - \hat{\boldsymbol{z}}_{m,k|k-1}) \tag{23}$$

Thus, the verification of the equivalence between Eqs 9 and 19 is transformed into the verification of the equivalence between Eqs 22 and 23. It is obviously that Eqs 17 and 18 in [38] are similar to Eqs 22 and 23. The de-correlating filter is considered to be different from the CGAF in [38], which will be discussed further based on Eqs 22 and 23.

Introducing *Lemma* 1 in [35], $\hat{\boldsymbol{z}}_{m,k|k}$ can be expressed as follows

$$\hat{\boldsymbol{z}}_{m,k|k} = \hat{\boldsymbol{z}}_{m,k|k-1} + \mathrm{E}(\boldsymbol{\Lambda}_{m,k}\boldsymbol{\Lambda}_{m,k}^{\mathrm{T}}|\boldsymbol{Z}_{m,k-1})(\boldsymbol{P}_{m,k|k-1}^{zz})^{-1}(\boldsymbol{z}_{m,k} - \hat{\boldsymbol{z}}_{m,k|k-1}) \tag{24}$$

where $\boldsymbol{\Lambda}_{m,k} = h_{m,k}(\boldsymbol{x}_k) - \mathrm{E}[h_{m,k}(\boldsymbol{x}_k)|\boldsymbol{Z}_{m,k-1}]$.

Substituting Eq 24 into Eq 22, we have

$$\hat{\boldsymbol{x}}_{m,k+1|k} = \int_{\mathbf{R}^n} f_k(\boldsymbol{x}_k)\mathrm{N}(\boldsymbol{x}_k;\hat{\boldsymbol{x}}_{m,\,k|k},\boldsymbol{P}_{m,\,k|k})\mathrm{d}\boldsymbol{x}_k + \boldsymbol{D}_{m,k}\boldsymbol{R}_{m,k}^{-1}\left(\boldsymbol{I} - \mathrm{E}(\boldsymbol{\Lambda}_{m,k}\boldsymbol{\Lambda}_{m,k}^{\mathrm{T}}|\boldsymbol{Z}_{m,k-1})(\boldsymbol{P}_{m,k|k-1}^{zz})^{-1}\right)(\boldsymbol{z}_{m,k}$$
$$- \hat{\boldsymbol{z}}_{m,k|k-1}) \tag{25}$$

where $\boldsymbol{I}$ is the identity matrix.

Insert $\mathrm{E}(\boldsymbol{\Lambda}_{m,k}\boldsymbol{\Lambda}_{m,k}^{\mathrm{T}}|\boldsymbol{Z}_{m,k-1}) = \boldsymbol{P}_{m,k|k-1}^{zz} - \boldsymbol{R}_{m,k}$ into Eq 25, we can obtain

$$\hat{\boldsymbol{x}}_{m,k+1|k} = \int_{\mathbf{R}^n} f_k(\boldsymbol{x}_k)\mathrm{N}(\boldsymbol{x}_k;\hat{\boldsymbol{x}}_{m,\,k|k},\boldsymbol{P}_{m,\,k|k})\mathrm{d}\boldsymbol{x}_k + \boldsymbol{D}_{m,k}(\boldsymbol{P}_{m,k|k-1}^{zz})^{-1}(\boldsymbol{z}_{m,k} - \hat{\boldsymbol{z}}_{m,k|k-1})$$
$$= \int_{\mathbf{R}^n} f_k(\boldsymbol{x}_k)\mathrm{N}(\boldsymbol{x}_k;\hat{\boldsymbol{x}}_{m,\,k|k},\boldsymbol{P}_{m,\,k|k})\mathrm{d}\boldsymbol{x}_k + \boldsymbol{D}_{m,k}\boldsymbol{R}_{m,k}^{-1}(\boldsymbol{z}_{m,k} - \hat{\boldsymbol{z}}_{m,k|k}) \tag{26}$$

Thus, Eq 22 is proved to be equivalent to Eq 23, that is to say, Eq 9 is equivalent to Eq 19.

According to the definition of $F_{m,k}(\boldsymbol{x}_k)$, Eq 10 is expanded as follows

$$\mathrm{Eq.}(10) = \mathrm{E}[\tilde{f}_{k|k}(\boldsymbol{x}_k)\tilde{f}_{k|k}^{\mathrm{T}}(\boldsymbol{x}_k)|\boldsymbol{Z}_{m,k}] - \boldsymbol{D}_{m,k}\boldsymbol{R}_{m,k}^{-1}\boldsymbol{\mu}^T - \boldsymbol{\mu}\boldsymbol{R}_{m,k}^{-1}\boldsymbol{D}_{m,k}^{\mathrm{T}} + \boldsymbol{Q}_{m,k} - \boldsymbol{D}_{m,k}(\boldsymbol{R}_{m,k}^{-1}$$
$$- \boldsymbol{R}_{m,k}^{-1}\boldsymbol{\kappa}\boldsymbol{R}_{m,k}^{-1})\boldsymbol{D}_{m,k}^{\mathrm{T}} \tag{27}$$

where $\boldsymbol{\mu} = \mathrm{E}[\tilde{f}_{k|k}(\boldsymbol{x}_k)\tilde{h}_{m,k|k}^{\mathrm{T}}(\boldsymbol{x}_k)|\boldsymbol{Z}_{m,k}]$, and $\boldsymbol{\kappa} = \mathrm{E}[\tilde{h}_{m,k|k}(\boldsymbol{x}_k)\tilde{h}_{m,k|k}^{\mathrm{T}}(\boldsymbol{x}_k)|\boldsymbol{Z}_{m,k}]$, with $\tilde{f}_{k|k}(\boldsymbol{x}_k) = f_k(\boldsymbol{x}_k) - \mathrm{E}[f_k(\boldsymbol{x}_k)|\boldsymbol{Z}_{m,k}]$, and $\tilde{h}_{m,k|k}(\boldsymbol{x}_k) = h_{m,k}(\boldsymbol{x}_k) - \mathrm{E}[h_{m,k}(\boldsymbol{x}_k)|\boldsymbol{Z}_{m,k}]$. Then $\boldsymbol{k}$ is linearized as follows

$$\boldsymbol{\kappa} = \boldsymbol{H}_{m,k}\mathrm{E}[\tilde{\boldsymbol{x}}_{m,k|k}\tilde{\boldsymbol{x}}_{m,k|k}^{\mathrm{T}}|\boldsymbol{Z}_{m,k}]\boldsymbol{H}_{m,k}^{\mathrm{T}}$$
$$= \boldsymbol{H}_{m,k}\boldsymbol{P}_{m,k|k}\boldsymbol{H}_{m,k}^{\mathrm{T}} \tag{28}$$

where $\boldsymbol{H}_{m,k} = \frac{\partial h_{m,k}(\boldsymbol{x}_k)}{\partial \boldsymbol{x}_k}|_{\boldsymbol{x}_k = \hat{\boldsymbol{x}}_{m,k|k}}, \tilde{\boldsymbol{x}}_{m,k|k} = \boldsymbol{x}_k - \hat{\boldsymbol{x}}_{m,k|k}$. Insert Eq 28 into Eq 27, we have

$$\mathrm{Eq.}(10) = \mathrm{E}[\tilde{f}_{k|k}(\boldsymbol{x}_k)\tilde{f}_{k|k}^{\mathrm{T}}(\boldsymbol{x}_k)|\boldsymbol{Z}_{m,k}] - \boldsymbol{D}_{m,k}\boldsymbol{R}_{m,k}^{-1}\boldsymbol{\mu}^T - \boldsymbol{\mu}\boldsymbol{R}_{m,k}^{-1}\boldsymbol{D}_{m,k}^{\mathrm{T}} + \boldsymbol{Q}_{m,k}$$
$$- \boldsymbol{D}_{m,k}(\boldsymbol{R}_{m,k}^{-1} - \boldsymbol{R}_{m,k}^{-1}\boldsymbol{H}_{m,k}\boldsymbol{P}_{m,k|k}\boldsymbol{H}_{m,k}^{\mathrm{T}}\boldsymbol{R}_{m,k}^{-1})\boldsymbol{D}_{m,k}^{\mathrm{T}} \tag{29}$$

According to Eq.(A.6) in [35], we can obtain $(\boldsymbol{P}_{m,k|k-1}^{zz})^{-1} = \boldsymbol{R}_{m,k}^{-1} - \boldsymbol{R}_{m,k}^{-1}\boldsymbol{H}_{m,k}\boldsymbol{P}_{m,k|k}\boldsymbol{H}_{m,k}^{\mathrm{T}}\boldsymbol{R}_{m,k}^{-1}$. So Eq 29 can be expressed as follows

$$\mathrm{Eq.}(10) = \mathrm{E}[\tilde{f}_{k|k}(\boldsymbol{x}_k)\tilde{f}_{k|k}^{\mathrm{T}}(\boldsymbol{x}_k)|\boldsymbol{Z}_{m,k}] - \boldsymbol{D}_{m,k}\boldsymbol{R}_{m,k}^{-1}\boldsymbol{\mu}^T - \boldsymbol{\mu}\boldsymbol{R}_{m,k}^{-1}\boldsymbol{D}_{m,k}^{\mathrm{T}} + \boldsymbol{Q}_{m,k} - \boldsymbol{D}_{m,k}(\boldsymbol{P}_{m,k|k-1}^{zz})^{-1}\boldsymbol{D}_{m,k}^{\mathrm{T}} \tag{30}$$

where the second and the third items from the right of equal sign satisfy

$(\boldsymbol{D}_{m,k}\boldsymbol{R}_{m,k}^{-1}\boldsymbol{\mu}^T)^T = \boldsymbol{\mu}\boldsymbol{R}_{m,k}^{-1}\boldsymbol{D}_{m,k}^{\mathrm{T}}$. Substituting Eqs 18 and 21 into Eq 20, one can obtain

$$
\begin{aligned}
\text{Eq.}(20) = \mathrm{E}[\tilde{f}_{k|k}(x_k)\tilde{f}_{k|k}^{\mathrm{T}}(x_k)|\boldsymbol{Z}_{m,k}] - \boldsymbol{D}_{m,k}(\boldsymbol{P}_{m,k+1|k}^{zz})^{-1}(\boldsymbol{P}_{m,k+1|k}^{xz})^{\mathrm{T}}\boldsymbol{P}_{m,k|k}^{-1}\boldsymbol{\gamma} \\
-\boldsymbol{\gamma}^{T}\boldsymbol{P}_{m,k|k}^{-1}\boldsymbol{P}_{m,k+1|k}^{xz}(\boldsymbol{P}_{m,k+1|k}^{zz})^{-1}\boldsymbol{D}_{m,k}^{\mathrm{T}} + \boldsymbol{Q}_{m,k} - \boldsymbol{D}_{m,k}(\boldsymbol{P}_{m,k|k-1}^{zz})^{-1}\boldsymbol{D}_{m,k}^{T}
\end{aligned}
\tag{31}
$$

where $\boldsymbol{\gamma} = \mathrm{E}[\tilde{\boldsymbol{x}}_{k|k}\tilde{f}_{k|k}^{\mathrm{T}}(\boldsymbol{x}_k)|\boldsymbol{Z}_{m,k}]$. The second and third terms on the right hand side are interchangeable.

By comparing Eqs 30 and 31, we can find that the second and third terms on the right side of the equal sign are different. Firstly, according to the definition of $\boldsymbol{\mu}$, the second item on the right of equal sign of Eq 30 is expanded as following

$$
\begin{aligned}
\boldsymbol{D}_{m,k}\boldsymbol{R}_{m,k}^{-1}\boldsymbol{\mu}^T &= \boldsymbol{D}_{m,k}\boldsymbol{R}_{m,k}^{-1}\{\mathrm{E}[\tilde{f}_{k|k}(\boldsymbol{x}_k)\tilde{h}_{m,k|k}^{\mathrm{T}}(\boldsymbol{x}_k)|\boldsymbol{Z}_{m,k}]\}^{\mathrm{T}} \\
&= \boldsymbol{D}_{m,k}\boldsymbol{R}_{m,k}^{-1}\mathrm{E}[h_{m,k|k}(\boldsymbol{x}_k)\tilde{f}_{k|k}^{\mathrm{T}}(\boldsymbol{x}_k)|\boldsymbol{Z}_{m,k}]
\end{aligned}
\tag{32}
$$

Substituting Eq 32 into Eq 30, one can get

$$
\begin{aligned}
\text{Eq.}(10) = \mathrm{E}[\tilde{f}_{k|k}(\boldsymbol{x}_k)\tilde{f}_{k|k}^{\mathrm{T}}(\boldsymbol{x}_k)|\boldsymbol{Z}_{m,k}] - \boldsymbol{D}_{m,k}\boldsymbol{R}_{m,k}^{-1}\mathrm{E}[h_{m,k|k}(\boldsymbol{x}_k)\tilde{f}_{k|k}^{\mathrm{T}}(\boldsymbol{x}_k)|\boldsymbol{Z}_{m,k}] \\
-\mathrm{E}[\tilde{f}_{k|k}(\boldsymbol{x}_k)h_{m,k|k}^{\mathrm{T}}(\boldsymbol{x}_k)|\boldsymbol{Z}_{m,k}]\boldsymbol{R}_{m,k}^{-1}\boldsymbol{D}_{m,k}^{\mathrm{T}} + \boldsymbol{Q}_{m,k} - \boldsymbol{D}_{m,k}(\boldsymbol{P}_{m,k|k-1}^{zz})^{-1}\boldsymbol{D}_{m,k}^{\mathrm{T}}
\end{aligned}
\tag{33}
$$

The second item on the right of equal sign in Eq 31 is expanded as follows

$$
\begin{aligned}
&\boldsymbol{D}_{m,k}(\boldsymbol{P}_{m,k+1|k}^{zz})^{-1}(\boldsymbol{P}_{m,k+1|k}^{xz})^{\mathrm{T}}\boldsymbol{P}_{m,k|k}^{-1}\boldsymbol{\gamma} \\
&= \boldsymbol{D}_{m,k}(\boldsymbol{P}_{m,k+1|k}^{zz})^{-1}(\boldsymbol{P}_{m,k+1|k}^{xz})^{\mathrm{T}}\boldsymbol{P}_{m,k|k}^{-1}\mathrm{E}[\tilde{\boldsymbol{x}}_{m,k|k}\tilde{f}_{k|k}^{\mathrm{T}}(\boldsymbol{x}_k)|\boldsymbol{Z}_{m,k}] \\
&= \boldsymbol{D}_{m,k}\boldsymbol{G}_{m,k+1}^{\mathrm{T}}\boldsymbol{P}_{m,k|k}^{-1}\mathrm{E}[\tilde{\boldsymbol{x}}_{m,k|k}\tilde{f}_{k|k}^{\mathrm{T}}(\boldsymbol{x}_k)|\boldsymbol{Z}_{m,k}]
\end{aligned}
\tag{34}
$$

where $\boldsymbol{G}_{m,k+1}^{\mathrm{T}}$ is the Kalman gain. According to the principle of the Kalman filter, we have $\boldsymbol{G}_{m,k+1} = \boldsymbol{P}_{m,k|k}\boldsymbol{H}_{m,k}^{\mathrm{T}}\boldsymbol{R}_{m,k}^{-1}$. Inserting it into Eq 34 yields

$$
\begin{aligned}
&\boldsymbol{D}_{m,k}(\boldsymbol{P}_{m,k+1|k}^{zz})^{-1}(\boldsymbol{P}_{m,k+1|k}^{xz})^{\mathrm{T}}\boldsymbol{P}_{m,k|k}^{-1}\boldsymbol{\gamma} \\
&= \boldsymbol{D}_{m,k}(\boldsymbol{P}_{m,k|k}\boldsymbol{H}_{m,k}^{\mathrm{T}}\boldsymbol{R}_{m,k}^{-1})^{T}\boldsymbol{P}_{m,k|k}^{-1}\mathrm{E}[\tilde{\boldsymbol{x}}_{m,k|k}\tilde{f}_{k|k}^{\mathrm{T}}(\boldsymbol{x}_k)|\boldsymbol{Z}_{m,k}] \\
&= \boldsymbol{D}_{m,k}\boldsymbol{R}_{m,k}^{-1}\boldsymbol{H}_{m,k}\mathrm{E}[\tilde{\boldsymbol{x}}_{m,k|k}\tilde{f}_{k|k}^{\mathrm{T}}(\boldsymbol{x}_k)|\boldsymbol{Z}_{m,k}] \\
&= \boldsymbol{D}_{m,k}\boldsymbol{R}_{m,k}^{-1}\mathrm{E}[h_{m,k|k}(\boldsymbol{x}_k)\tilde{f}_{k|k}^{\mathrm{T}}(\boldsymbol{x}_k)|\boldsymbol{Z}_{m,k}]
\end{aligned}
\tag{35}
$$

Consider the mutual transposition items in Eq 31, and substitute Eq 35 into Eq 31 to get

$$
\begin{aligned}
\text{Eq.}(20) = \mathrm{E}[\tilde{f}_{k|k}(x_k)\tilde{f}_{k|k}^{\mathrm{T}}(x_k)|\boldsymbol{Z}_{m,k}] - \boldsymbol{D}_{m,k}\boldsymbol{R}_{m,k}^{-1}\mathrm{E}[h_{m,k|k}(\boldsymbol{x}_k)\tilde{f}_{k|k}^{\mathrm{T}}(\boldsymbol{x}_k)|\boldsymbol{Z}_{m,k}] \\
-\mathrm{E}[\tilde{f}_{k|k}(\boldsymbol{x}_k)h_{m,k|k}^{\mathrm{T}}(\boldsymbol{x}_k)|\boldsymbol{Z}_{m,k}]\boldsymbol{R}_{m,k}^{-1}\boldsymbol{D}_{m,k}^{\mathrm{T}} + \boldsymbol{Q}_{m,k} - \boldsymbol{D}_{m,k}(\boldsymbol{P}_{m,k|k-1}^{zz})^{-1}(\boldsymbol{D}_{m,k})^{T} \\
= \text{Eq.}(10)
\end{aligned}
\tag{36}
$$

Thus, Eq 20 is proved to be equivalent to Eq 10. Eqs 22–36 show that AFF1-CN is equivalent to AFF2-CN. The equivalence between the GASF and de-correlating filters has been proved, and it can be further inferred that AFF1-CN and AFF2-CN is also equivalent to HCFF-CN, since all three are approximated by the five-degree cubature rule.

## 5. Simulations and analysis

In this section, maneuvering target tracking simulations are performed to test the effectiveness of the proposed filters, compared with the traditional GFF and HCFF-CN. The Gaussian

weighted integrals of all filters are approximated by the fifth-degree cubature rule in [41]. Furthermore, the proposed filters and SGFS-CN are compared to analyze the positive effect of fusion on AFFs-CN performance.

A classic maneuvering target tracking problem is considered, which performs maneuvering turns on the horizontal plane at a constant turning rate [40, 41]. The turning motion and measurement model can be generalized as follows

$$
\begin{cases}
\boldsymbol{x}_{k+1} = \begin{bmatrix} 1 & \sin\Omega\Delta t/\Omega & 0 & -(1-\cos\Omega\Delta t)/\Omega \\ 0 & \cos\Omega\Delta t & 0 & -\sin\Omega\Delta t \\ 0 & (1-\cos\Omega\Delta t)/\Omega & 1 & \sin\Omega\Delta t/\Omega \\ 0 & \sin\Omega\Delta t & 0 & \cos\Omega\Delta t \end{bmatrix} \boldsymbol{x}_k + \boldsymbol{w}_k \\[4em]
\boldsymbol{z}_{m,k} = \begin{bmatrix} \sqrt{(\xi_k - \xi_{m,r})^2 + (\varsigma_k - \varsigma_{m,r})^2} \\[1em] \tan^{-1}\left(\dfrac{\varsigma_k - \varsigma_{m,r}}{\xi_k - \xi_{m,r}}\right) \end{bmatrix} + \boldsymbol{v}_{m,k}
\end{cases}
\tag{37}
$$

where $\boldsymbol{x}_k = \begin{bmatrix} \xi_k & \dot{\xi}_k & \varsigma_k & \dot{\varsigma}_k \end{bmatrix}^{\mathrm{T}}$, $\xi_k$ and $\varsigma_k$ denote the target positions in X and Y directions; $\dot{\xi}_k$ and $\dot{\varsigma}_k$ indicate the velocities in the X and Y directions respectively. $\xi_{m,r}$ and $\varsigma_{m,r}$ stand for the *mth* radar sensor positions in X and Y directions. $\Omega$ is a known and constant turning rate. $\Delta t$ is the time interval between two consecutive measurements. The definition of process noise $\boldsymbol{w}_k$ and observation noise $\boldsymbol{v}_{m,k}$ is the same as Eq 1, where $\boldsymbol{Q}_k$ satisfies

$$
\boldsymbol{Q}_k = \mathrm{E}[\boldsymbol{w}_k \boldsymbol{w}_k^{\mathrm{T}}] = \begin{bmatrix} \Delta t^3/3 & \Delta t^2/2 & 0 & 0 \\ \Delta t^2/2 & \Delta t & 0 & 0 \\ 0 & 0 & \Delta t^3/3 & \Delta t^2/2 \\ 0 & 0 & \Delta t^2/2 & \Delta t \end{bmatrix}
\tag{38}
$$

The true initial states together with its associated covariance are defined as follows

$$
\begin{aligned}
\boldsymbol{x}_0 &= \begin{bmatrix} 1000m & 300ms^{-1} & 1000m & 0ms^{-1} \end{bmatrix}^{\mathrm{T}} \\
\boldsymbol{P}_{0|0} &= \mathrm{diag}\left(\begin{bmatrix} 100m^2 & 10m^2s^{-2} & 100m^2 & 10m^2s^{-2} \end{bmatrix}\right) \\
\Delta t &= 1s, \quad \Omega = -3°s^{-1}
\end{aligned}
\tag{39}
$$

where $\hat{\boldsymbol{x}}_{0|0} \sim \mathrm{N}(\boldsymbol{x}_0, \boldsymbol{P}_{0|0})$. Two radars are used as tracking sensors; Fig 1 indicates the fixed radars positions and the true trajectory throughout 10 sample times.

For fair comparisons, independent Monte Carlo tests run 50 ($L$) times. The total number of scans per run is 100. All the filters are initialized in the same way each run. To compare different nonlinear filters' performance, the metric are defined to be the root mean square error (RMSE). For example, the RMSE in position at time $k+1$ is defined as

$$
RMSE_{k+1}^{pos} = \sqrt{\frac{1}{L}\sum_{n=1}^{L}\left((\xi_{k+1} - \hat{\xi}_{k+1}^n)^2 + (\varsigma_{k+1} - \hat{\varsigma}_{k+1}^n)^2\right)}
\tag{40}
$$

where $(\xi_{k+1}, \zeta_{k+1})$ is the true position at time $k+1$, and $(\hat{\xi}_{k+1}^n, \hat{\varsigma}_{k+1}^n)$ is the estimated position at $k+1$ from the *nth* Monte Carlo run. The RMSE in velocity can be obtained in the way as the RMSE in position.

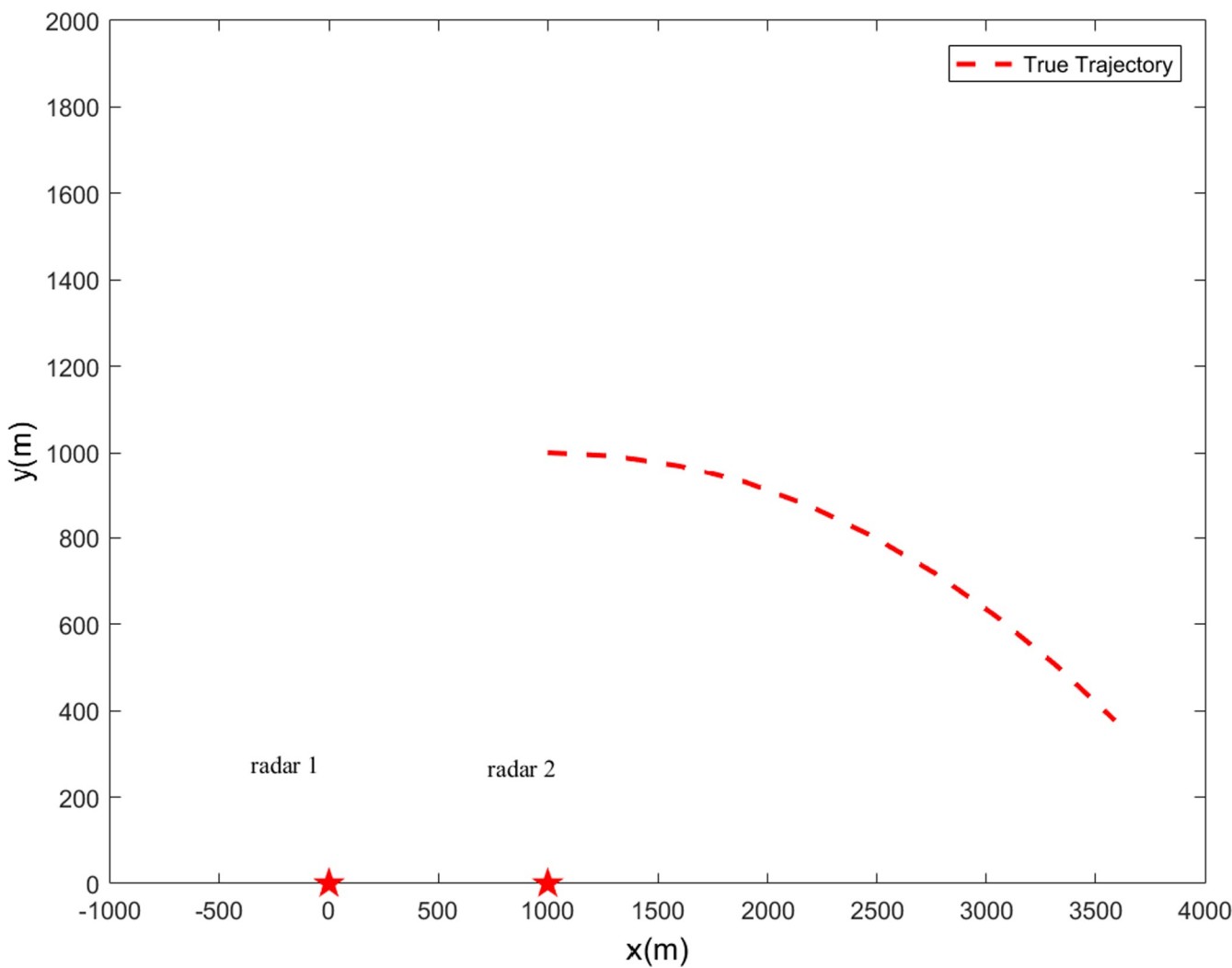

**Fig 1. Target true trajectory (★-radars' positions).**

(1) Scenario 1

In Scenario 1, let $D_{m,k} = 0$, which implies that $w_k$ and $v_{m,k}$ are uncorrelated with each other. The covariance of $v_{m,k}$ satisfies $R_{m,k} = \text{diag}[1600m^2 \quad 200mrad^2]$. The RMSE results in the position and velocity for Scenario 1 are indicated in Fig 2, where the SGF type, including traditional GF, de-correlating filter, CGAF, and GASF, only uses radar 1 for target tracking.

As shown in Fig 2A, the position and velocity RMSE curves of each Gaussian filter involved almost coincide. That is to say, in this case, they have almost the same tracking accuracy. Therefore, the research results in [34–38] are considered to be repeated. In addition, by comparing RMSE curves in Fig 2A and 2B, it can be seen that AFF1-CN and AFF2-CN can improve the tracking accuracy of corresponding de-correlation filter and CGAF. It can be deduced that AFF1-CN and AFF2-CN can fuse the tracking data of Radar 1 and Radar 2, rather than just a radar sensor. Comparing the results in Fig 2B, it can be seen that the RMSE curves of AFF1-CN, AFF2-CN, GFF and HCFF-CN are very close, which is supported by the results in Table 1. Therefore, it can be concluded that in the case of no correlation between

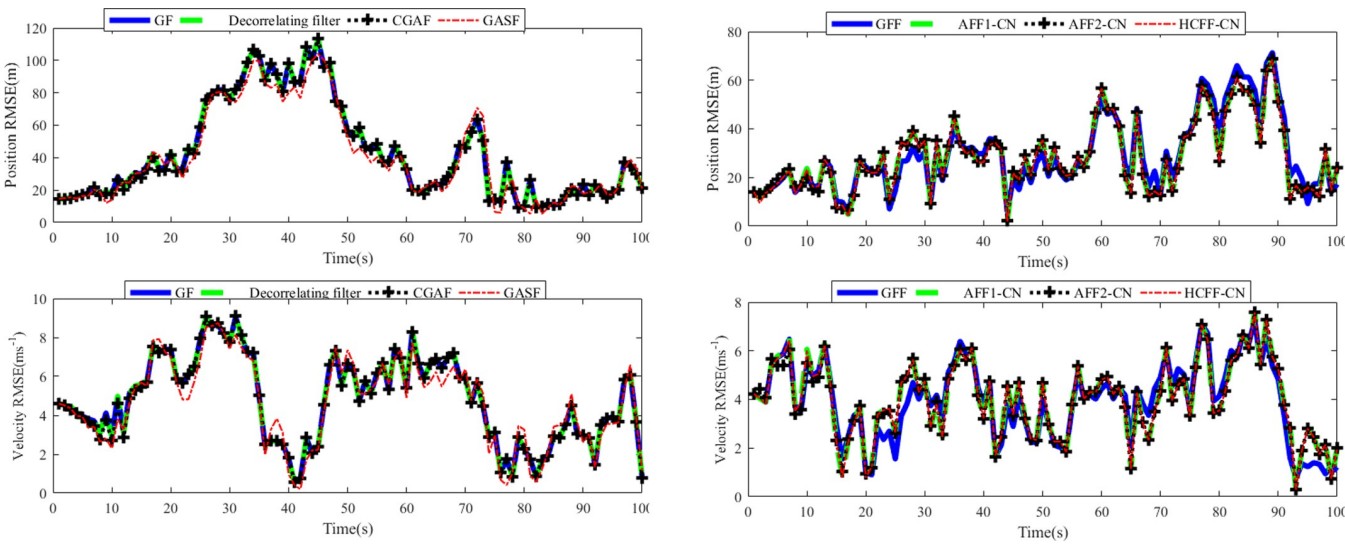

**Fig 2.** RMSEs of position and velocity in scenario 1: (a) single filter; (b) fusion filter.

process noise and measurement noise, the performance of the two filters proposed is almost as good as the traditional GFF and HCFF-CN filters, and better than the SGFs-CN filters.

(2) Scenario 2

In Scenario 2, let $\boldsymbol{v}_{m,k} = \boldsymbol{b}_m\boldsymbol{w}_k$, $\boldsymbol{R}_{m,k} = \boldsymbol{b}_m\boldsymbol{Q}_k\boldsymbol{b}_m^{\mathrm{T}}$, and $\boldsymbol{D}_{m,k} = \boldsymbol{Q}_k\boldsymbol{b}_m^{T} \neq 0$ [27]. The correlation coefficient $\boldsymbol{b}_m$ = [0.06 0.06 0.006 0.006; 0.006 0.006 0.006 0.006]. $\boldsymbol{D}_{m,k} \neq 0$ implies that $\boldsymbol{w}_k$ and $\boldsymbol{v}_{m,k}$ are cross-correlated. RMSE results of position and speed within the range of 10s-100s are shown in Fig 3 and Table 2. In Fig 3A, traditional GF and SGFs-CN still only use Radar 1 for target tracking.

It can be inferred from Fig 3A that when the process and measurement noises are cross-correlated, the de-correlating filter, CGAF, and GASF can improve the accuracy of the traditional GF. By comparing Fig 3A and 3B, it can be seen that the tracking accuracy of AFF1-CN and AFF2-CN remains superior to the corresponding SGF types for the same reason as in Scenario 1. As shown in Fig 3B, AFF1-CN and AFF2-CN can achieve the better position and velocity accuracy than traditional GFF. In addition, the position and velocity tracking accuracy of AFF1-CN, AFF2-CN, and HCFF-CN keep very close, which is also supported by the result in Table 2. The AFF2-CN has the best position tracking accuracy and AFF1-CN has the best velocity tracking accuracy. Therefore, the proposed AFF1-CN and AFF2-CN achieve better tracking performance than the traditional GFF and SGFs, and similar performance to HCFF-CN.

(3) Scenario 3

Considering in a practical multi-radar tracking system, the cross-correlations between each local measurement noise and process noise are different. The correlation coefficients $\boldsymbol{b}_m$ of both radars are set as $\boldsymbol{b}_1/0.006$ = [10 10 1 1; 1 1 1 1] and $\boldsymbol{b}_2/0.003$ = [10 10 10 1; 1 1 10 1]

**Table 1. RMSEs of position and velocity in scenario 1.**

| Algorithms | GFF | AFF1-CN | AFF2-CN | HCFF-CN |
|---|---|---|---|---|
| The Mean of Position RMSEs (m) | 29.2037 | 28.5804 | 28.6522 | 28.4419 |
| The Mean of Velocity RMSEs (m/s) | 3.9461 | 4.0231 | 4.0052 | 4.0103 |

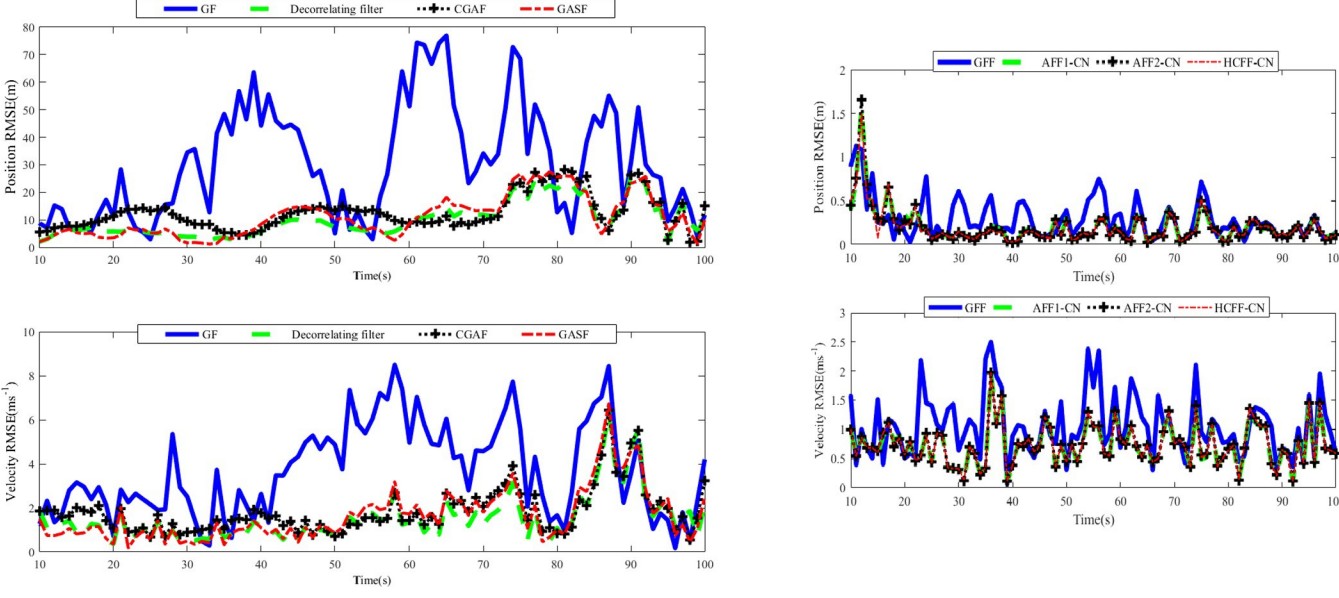

**Fig 3.** RMSEs of position and velocity in scenario 2: (a) single filter; (b) fusion filter.

respectively. Accordingly, the $\boldsymbol{v}_{m,k}$, $\boldsymbol{R}_{m,k}$ and $\boldsymbol{D}_{m,k}$ in each local filter are also different. In this Scenario, only the performance of GFF, AFFs-CN and HCFF-CN are compared. The position and velocity RMSE results are shown in Fig 4.

Judging from Fig 4 and Table 3, the position and speed tracking performance of the four filters is very good, among which GFF tracking RMSE is still slightly higher than other filters. The RMSE results of AFFs-CN and HCFF-CN are hard to distinguish just as in Scenario 2. Only the data in Table 3 show some differences between the two. Therefore, the same conclusion can be drawn as Scenario 2.

**Remark 1**: *Although the above results show that AFF1-CN, AFF2-CN and HCFF-CN are equivalent, the assumptions of the three filters are different. So we can choose the most appropriate filter according to the practical initial conditions.*

## 6. Conclusion

To adapt the traditional FF to the nonlinear discrete dynamic stochastic system with cross-correlative noises, two AFFs-CN are proposed based on the de-correlating filter and CGAF. These two filters are suitable for both white noise independent system and noise cross-correlation system, and the theoretical equivalence of the two algorithms in the nonlinear fusion system has been verified. The simulation results show that AFFs-CN have almost the same performance as GFF and HCFF-CN, and AFFs-CN achieve better performance than SGF in the case of no correlation between measuring noise and dealing with noise, and AFFs-CN have superior accuracy and robustness than GFF and SGFs when the measurement noise and process

**Table 2. RMSEs of position and velocity in scenario 2 (10-100s).**

| Algorithms | GFF | AFF1-CN | AFF2-CN | HCFF-CN |
|---|---|---|---|---|
| The Mean of Position RMSEs (m) | 0.2873 | 0.2027 | 0.1943 | 0.1984 |
| The Mean of Velocity RMSEs (m/s) | 1.0570 | 0.7307 | 0.7349 | 0.7385 |

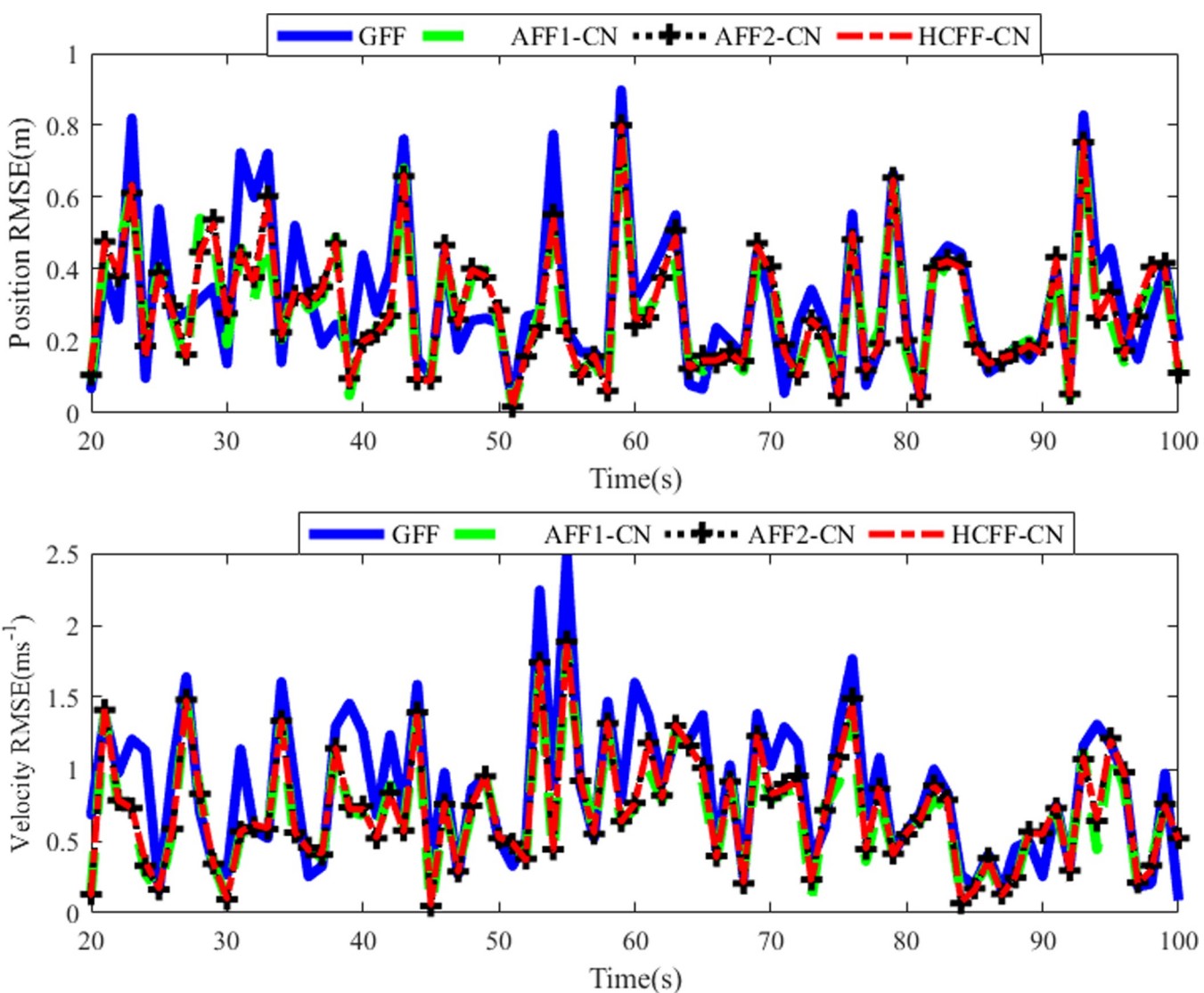

**Fig 4. RMSEs of position and velocity in scenario 3.**

**Table 3. RMSEs of position and velocity in scenario 3 (20-100s).**

| Algorithms | GFF | AFF1-CN | AFF2-CN | HCFF-CN |
|---|---|---|---|---|
| The Mean of Position RMSEs (m) | 0.3605 | 0.3295 | 0.3352 | 0.3360 |
| The Mean of Velocity RMSEs (m/s) | 0.8989 | 0.6787 | 0.7084 | 0.7085 |

noise are cross-correlated. The future work will focus on the nonlinear federated filter with auto-correlated noises.

## Supporting information

**S1 File. Minimal dataset.**
(ZIP)

## Acknowledgments

The authors gratefully acknowledge the helpful comments and suggestions of the reviewers, which have improved the presentation.

## Author Contributions

**Conceptualization:** Sisi Wang.

**Formal analysis:** Sisi Wang.

**Funding acquisition:** Lijun Wang, Sisi Wang.

**Investigation:** Sisi Wang.

**Methodology:** Lijun Wang, Sisi Wang.

**Software:** Sisi Wang.

**Writing – original draft:** Lijun Wang, Sisi Wang, Wenzhi Yang.

**Writing – review & editing:** Wenzhi Yang.

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
