## [Decision Letter · Decision Letter 0]

21 Oct 2020

PONE-D-20-25635

Adaptive Federated Filter for Multi-sensor Nonlinear System with Cross-correlated Noises

PLOS ONE

Dear Dr. Wang,

Thank you for submitting your manuscript to PLOS ONE. After careful consideration, we feel that it has merit but does not fully meet PLOS ONE’s publication criteria as it currently stands. Therefore, we invite you to submit a revised version of the manuscript that addresses the points raised during the review process.

ACADEMIC EDITOR: Based on the two reviewers' comments, the paper needs to be further improved and completed before publication. Please very carefully revise it in the next round. Moreover, the switched systems approach is also a very popular intelligent control strategy, and hence some outlooks about the current work can be provided combining with the latest results on switched control issue, such as Multiple Lyapunov Functions Analysis Approach for Discrete-Time Switched Piecewise-Affine Systems Under Dwell-Time Constraints; Observer-Based Control for Cyber-Physical Systems With DoS Attacks Via A Cyclic Switching Strategy.

We look forward to receiving your revised manuscript.

Kind regards,

Academic Editor

PLOS ONE

Journal Requirements:

"The work was supported in part by Fund of Hubei Key Laboratory of Inland Shipping Technology under Grant (NHHY2018003); Scientific Research Start-up Funds of Guangdong Ocean University under Grants (E15031, R17012); Characteristic Innovation Projects of Guangdong Province under Grants (2017KTSCX088, 2017KTSCX092, 2019KTSCX230).".

i) Please provide an amended statement that declares *all* the funding or sources of support (whether external or internal to your organization) received during this study, as detailed online in our guide for authors at http://journals.plos.org/plosone/s/submit-now.  Please also include the statement “There was no additional external funding received for this study.” in your updated Funding Statement.

ii) Please include your amended Funding Statement within your cover letter. We will change the online submission form on your behalf.

Additional Editor Comments (if provided):

Based on the two reviewers' comments, the paper needs to be further improved and completed before publication. Please very carefully revise it in the next round. Moreover, the switched systems approach is also a very popular intelligent control strategy, and hence some outlooks about the current work can be provided combining with the latest results on switched control issue, such as Multiple Lyapunov Functions Analysis Approach for Discrete-Time Switched Piecewise-Affine Systems Under Dwell-Time Constraints; Observer-Based Control for Cyber-Physical Systems With DoS Attacks Via A Cyclic Switching Strategy.

Reviewers' comments:

Reviewer's Responses to Questions

**Comments to the Author**

1. Is the manuscript technically sound, and do the data support the conclusions?

Reviewer #1: Yes

Reviewer #2: Yes

2. Has the statistical analysis been performed appropriately and rigorously? 

Reviewer #1: Yes

Reviewer #2: Yes

3. Have the authors made all data underlying the findings in their manuscript fully available?

Reviewer #1: Yes

Reviewer #2: Yes

4. Is the manuscript presented in an intelligible fashion and written in standard English?

Reviewer #1: Yes

Reviewer #2: Yes

5. Review Comments to the Author

Reviewer #1: This paper proposes two adaptive federated filters for nonlinear discrete dynamic stochastic system with cross-correlative noises. These two filters are suitable for both white noise independent system and noise cross-correlation system. The contribution of the paper is clear. This paper can be published after some minor modifications.

1. The aim of this paper is to design adaptive filtering methods for systems with cross-correlative noises, but some related works are missing, such as “A novel outlier-robust kalman filtering framework based on statistical similarity measure”, “A novel robust Student’s t-based Kalman filter”, “A new adaptive extended Kalman filter for cooperative localization”, “A novel robust Gaussian–Student’s t mixture distribution based Kalman filter”, “A novel Kullback–Leibler divergence minimization-based adaptive student’s t-filter”, “Variational adaptive Kalman filter with Gaussian-inverse-Wishart mixture distribution”, “A slide window variational adaptive Kalman filter”, “Design of Gaussian approximate filter and smoother for nonlinear systems with correlated noises at one epoch apart”.

2. Please explain that how to get the additional term in Eq (7)?

3. What are the technical challenges in devising these presented filters?

4. Please explain the differences between adaptive federated filter 1 for cross-correlated noises (AFF1-CN), adaptive federated filter 2 for cross-correlated noises(AFF2-CN) and high degree cubature federated filter for cross-correlated noises(HCFF-CN) essentially, and point out the advantages of AFF1-CN and AFF2-CN over HCFF-CN.

5. What is the meaning of P_{m} which lies in line 94 in this paper ? Please check and correct it.

Reviewer #2: This paper presents an adaptive approach to the federated filter for multi-sensor nonlinear systems with cross-correlations between process noise and local measurement noise. Simulation has validated that the proposed algorithms are superior to the traditional federated filter and Gaussian filter with same-paced correlated noises. The paper is well written and organized reasonably. However, it still suffers from the following problems to be addressed.

1.The research significance of this paper should be further strengthened by combining with the practical application of cross-correlations between process noise and local measurement noise.

2.Eq. (38) should be not the cross-correlated covariance.

3.A practical example should be provided to validate the proposed method in practical application and illustrate the research significance.

4.The literature review on nonlinear filtering and data fusion should be improved by considering the following references:

[1]Multi-sensor optimal data fusion for INS/GNSS/CNS integration based on unscented Kalman filter. International Journal of Control, Automation and Systems 2018; 16(1): 129-140.

[2]Unscented Kalman Filter with Process Noise Covariance Estimation for Vehicular INS/GPS Integration System. Information Fusion, 2020, 64: 194-204

[3]Maximum likelihood principle and moving horizon estimation based adaptive unscented Kalman filter. Aerospace Science and Technology 2018; 73: 184-196.

[4]Model predictive based unscented Kalman filter for hypersonic vehicle navigation with INS/GNSS integration. IEEE Access 2020; 8: 4814 - 4823.

Overall, this manuscript can be accepted by the Journal “PLOS ONE” after the revision.

6. PLOS authors have the option to publish the peer review history of their article (what does this mean?). If published, this will include your full peer review and any attached files.

Reviewer #1: No

Reviewer #2: No

---

## [Author Response · Author response to Decision Letter 0]

9 Dec 2020

All review comments have been answered, please refer to the specific document 'Response to Reviewers'.

Many thanks to the reviewers and editors for their valuable comments.

---

## [Decision Letter · Decision Letter 1]

25 Jan 2021

Adaptive Federated Filter for Multi-sensor Nonlinear System with Cross-correlated Noises

PONE-D-20-25635R1

Dear Dr. Wang,

We’re pleased to inform you that your manuscript has been judged scientifically suitable for publication and will be formally accepted for publication once it meets all outstanding technical requirements.

Kind regards,

Yanzheng Zhu

Academic Editor

PLOS ONE

Additional Editor Comments:

Based on the reviewers' comments, the paper can be accepted for publication now. Moreover, some outlooks about the current work can be provided combining with the latest results in the following on switched control issue. It is better to discuss them in the conclusion before submitting the final version.

[1] Quasi-Synchronization of Discrete-Time Lur’e-Type Switched Systems with Parameter Mismatches and Relaxed PDT Constraints, IEEE Transactions on Cybernetics, 2020, 50(5): 2026-2037.

[2] Multiple Lyapunov Functions Analysis Approach for Discrete-Time Switched Piecewise-Affine Systems Under Dwell-Time Constraints, IEEE Transactions on Automatic Control, 2020, 65(5): 2177-2184.

Reviewers' comments:

Reviewer's Responses to Questions

**Comments to the Author**

1. If the authors have adequately addressed your comments raised in a previous round of review and you feel that this manuscript is now acceptable for publication, you may indicate that here to bypass the “Comments to the Author” section, enter your conflict of interest statement in the “Confidential to Editor” section, and submit your "Accept" recommendation.

Reviewer #1: All comments have been addressed

Reviewer #2: All comments have been addressed

2. Is the manuscript technically sound, and do the data support the conclusions?

Reviewer #1: Yes

Reviewer #2: Yes

3. Has the statistical analysis been performed appropriately and rigorously? 

Reviewer #1: Yes

Reviewer #2: Yes

4. Have the authors made all data underlying the findings in their manuscript fully available?

Reviewer #1: Yes

Reviewer #2: Yes

5. Is the manuscript presented in an intelligible fashion and written in standard English?

Reviewer #1: Yes

Reviewer #2: Yes

6. Review Comments to the Author

Reviewer #1: The authors have addressed the reviewer's comments well. So, the reviewer recommends to accept this paper for publication.

Reviewer #2: No new critical comments from my side. The authors have improved the revised version as per all the Reviewers' suggestions

7. PLOS authors have the option to publish the peer review history of their article (what does this mean?). If published, this will include your full peer review and any attached files.

Reviewer #1: No

Reviewer #2: No

---

## [Editor Report · Acceptance letter]

4 Feb 2021

PONE-D-20-25635R1 

Adaptive federated filter for multi-sensor nonlinear system with cross-correlated noises 

Dear Dr. Wang:

I'm pleased to inform you that your manuscript has been deemed suitable for publication in PLOS ONE. Congratulations! Your manuscript is now with our production department. 

Kind regards, 

on behalf of

Dr. Yanzheng Zhu 

Academic Editor

PLOS ONE